# Drivers of respiratory syncytial virus seasonal epidemics in children under 5 years in Kilifi, coastal Kenya

James Wambua[1¤]*, Patrick K. Munywoki[1], Pietro Coletti[2], Bryan O. Nyawanda[3], Nickson Murunga[1], D. James Nokes[1,4], Niel Hens[2,5]

**1** Kenya Medical Research Institute, KEMRI -Wellcome Trust Research Programme (KWTRP), Kilifi, Kenya, **2** Data Science Institute, I-BioStat, Hasselt University, Hasselt, Belgium, **3** Kenya Medical Research Institute, Center for Global Health Research, Kisumu, Kenya, **4** School of Life Sciences, University of Warwick, Coventry, United Kingdom, **5** Centre for Health Economics Research and Modelling Infectious Diseases, Vaccine & Infectious Disease Institute, University of Antwerp, Antwerp, Belgium

¤ Current address: Data Science Institute, I-BioStat, Hasselt University, Hasselt, Belgium
* james.wambua@uhasselt.be

**Data Availability Statement:** The data underlying the results presented in the study are available from Harvard Dataverse database (https://doi.org/10.7910/DVN/TQ4EXK).

## Abstract

Respiratory syncytial virus (RSV) causes significant childhood morbidity and mortality in the developing world. The determinants of RSV seasonality are of importance in designing interventions. They are poorly understood in tropical and sub-tropical regions in low- and middle-income countries. Our study utilized long-term surveillance data on cases of RSV associated with severe or very severe pneumonia in children aged 1 day to 59 months admitted to the Kilifi County Hospital. A generalized additive model was used to investigate the association between RSV admissions and meteorological variables (maximum temperature, rainfall, absolute humidity); weekly number of births within the catchment population; and school term dates. Furthermore, a time-series-susceptible-infected-recovered (TSIR) model was used to reconstruct an empirical transmission rate which was used as a dependent variable in linear regression and generalized additive models with meteorological variables and school term dates. Maximum temperature, absolute humidity, and weekly number of births were significantly associated with RSV activity in the generalized additive model. Results from the TSIR model indicated that maximum temperature and absolute humidity were significant factors. Rainfall and school term did not yield significant relationships. Our study indicates that meteorological parameters and weekly number of births potentially play a role in the RSV seasonality in this region. More research is required to explore the underlying mechanisms underpinning the observed relationships.

## Introduction

Respiratory syncytial virus (RSV) is an ubiquitous RNA virus that is a significant cause of early childhood morbidity and mortality worldwide [1,2]. In low- and middle-income settings, RSV is recognized as an important cause of hospitalized severe pneumonia in children aged

**Funding:** JN received funding from Wellcome Trust (grant number 102975). JW received a travel master's fellowship from VLIR-UOS. The funders had no role in study design, data collection and analysis, decision to publish, or preparation of the manuscript.

**Competing interests:** The authors have declared that no competing interests exist.

between 28 days and 5 years [1]. In 2015, it was estimated that there were 33.1 million (uncertainty range 21.6–50.3) RSV- Acute Lower Respiratory Infections (ALRI) episodes globally, leading to about 3.2 (2.7–3.8) million hospital admissions, and 59,600 (48,000–74,500) hospital deaths, in children < 5 years of age [2]. An approximate 1.4 million (1.2–1.7) in hospital admissions, and 27,300 (20,700–36,200) in-hospital deaths were in young children < 6 months of age. The vast majority of both morbidity and mortality arises from low- and middle-income countries [2].

There is no recognised prophylactic for RSV-ALRI for general use. After over 50 years of vaccine research and development, there is none, to date, that has been licensed [3]. Several RSV vaccines and monoclonal antibody products are at various stages of pre-clinical and clinical development targeting infants and pregnant women [4]. Most recently, a prophylactic high-potency long half-life anti-F protein monoclonal antibody met its primary endpoint in a Phase III trial [5]. Two maternal booster F protein vaccines are in Phase III trials [6]. The successful use of these products in early infants and pregnant women to prevent severe disease in children in the first few months of life will be affected by the timing of seasonal RSV transmission [7].

RSV is characterized by marked annual or biennial seasonal patterns [8]. In temperate climates, RSV infections have been found to happen in the winter months [9,10]. In tropical regions, a wide variation in the timing has been observed [10]. However, comparatively little attention has been directed to most tropical settings in Africa, hence the RSV seasonality pattern remains largely unexplored in these regions, despite the considerable ALRI burden and deaths [10,11]. To understand RSV seasonality, factors that drive these patterns should be well understood. In regions with temperate climates, RSV seasonal epidemics peak in winter months [11]. Previous studies in different tropical regions worldwide have yielded inconsistent results. Temperature, humidity and precipitation have been found to be strongly, weakly or not at all associated with RSV activity [12–18]. Seasonality in births has also been shown to impact the seasonality in infectious diseases, especially with respect to first infections [19,20]. Malnutrition has been found to be associated with RSV seasonality [21,22]. Demographic factors such as household crowding indices have been found to influence the length of RSV seasons [23].

Seasonal patterns of RSV in Kenya, through which the equator passes have been compared in 3 distinct counties (Nairobi, Siaya and Kilifi) [24]. However, the potential factors driving these seasonal patterns have not been assessed. The current study was carried out using previously collected datasets from the Kenya Medical Research Institute (KEMRI)-Wellcome Trust Research Programme (KWTRP), located in Kilifi Town along the coastal region bordering the Indian Ocean. The study aimed to investigate whether meteorological parameters (maximum temperature, absolute humidity, rainfall), weekly number of births, and school terms could be determinants of seasonal RSV epidemics. Identifying the potential factors driving RSV seasonality would enhance the understanding of the virus transmission mechanisms in this setting.

## Methods

### Ethics statement

The ethical approval for the study was granted by the Kenya Medical Research Institute–Scientific Ethics Review Unit (reference number SERU 3178). Written informed consent was obtained from parents or guardians for all eligible participants before any specimen(s) was collected for RSV. The data was analysed anonymously.

### Inclusivity in global research

Additional information regarding the ethical, cultural, and scientific considerations specific to inclusivity in global research is included in the Supporting Information (S1 Checklist).

### Data

Data used were collected from routine longitudinal hospital case surveillance for RSV, and routine longitudinal surveillance for the number of births and meteorological variables. The datasets are described below.

### RSV data

RSV data was obtained from Kilifi County Hospital located in Kilifi Town. Data was collected from children aged 1 day to 59 months who presented with a clinical syndrome for either severe or very severe pneumonia [25]. Written informed consent was obtained from parents or guardians for all eligible participants before any specimen(s) was collected. The data spanned the years 2002 to 2018. RSV was screened through immunofluorescence antibody test (IFAT) for all years and real time reverse transcription-polymerase chain reaction (RT-rtPCR) from 2007 [25]. Only data from locations covered by the Kilifi Health and Demographic Surveillance System (KHDSS) were included [26]. The KHDSS area extends roughly 40km north and south of Kilifi town and 30km in land, largely rural agricultural and the population size has approximately increased from 200,000 to 300,000 over the study period (Fig 1).

### Births data

Weekly number of birth data was obtained from the KHDSS registers. The system records the number of births, deaths, pregnancies and migration events. The data spanned the years 2002 to 2018.

### Meteorological data

Meteorological data was obtained from two weather stations. One station at Pwani University, Kilifi Town and the other at KWTRP. Data from the Pwani station was recorded manually and consisted of daily measurements of four main variables; maximum/minimum temperature (in degree Celsius), rainfall (in millimeters), and average relative humidity (expressed in %). The data comprised of measurements from January 2002 to July 2014. Data from KWTRP station for maximum/minimum temperature, rainfall, and average relative humidity was measured by an automatic weather station (Synoptic Automatic Weather Stations, Sutron, USA). Data from this station was collected from July 2010 to November 2018. Absolute humidity was obtained using equation 1, see supplementary material.

### School term data

The data consisted of school start dates and end dates of half terms and holidays for pre-primary and primary [28]. Children attend pre-primary schooling for 2–3 years and encompasses the age group 3–5 years. Primary school is attended for 8 years and includes children 6 years and above.

### Data processing

Weekly averages of meteorological variables were used in the analysis. Due to missing data in the two meteorological datasets, multiple imputation using multivariate imputation by chained

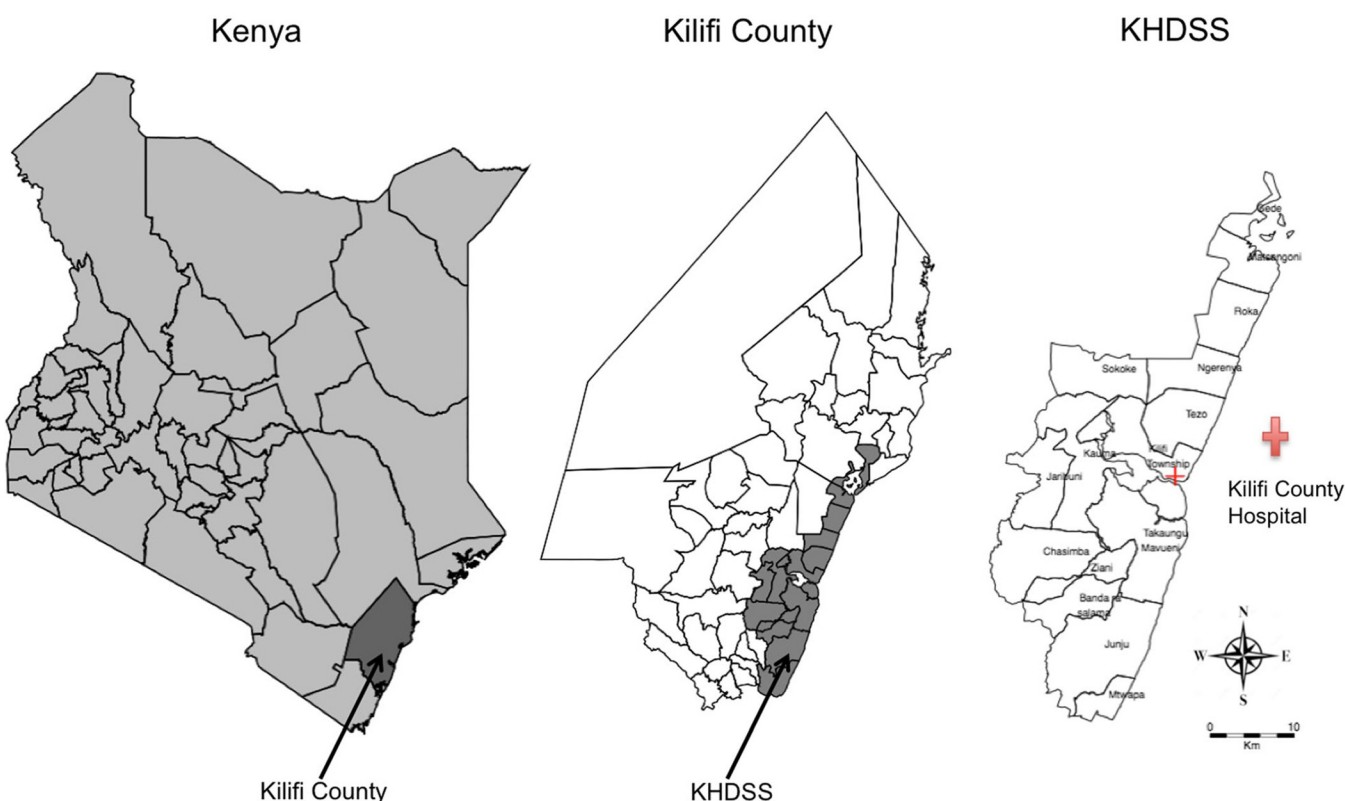

**Fig 1. Geographical location of the study area.** Left: Location of Kilifi county in Kenya. Middle: Map of Kilifi county showing KHDSS boundaries. Right: KHDSS map showing Kilifi County Hospital location. Reprinted from [27] under a CC BY license, with permission from BioMed Central, Alice Kamau (original copyright owner) [2017].

equations (MICE package) implemented in the R Statistical software (version 4.1.1) was performed on the daily observations [29]. We performed five imputations resulting in five datasets for each meteorological dataset. The number of imputations (5) was chosen based on imputation guidelines by Rubin [30] on the percentages of missing data. The resulting datasets were merged with the RSV data, birth data, and data on school term by year and calendar week. We performed separate analyses for the two meteorological datasets. See supplementary material for more details on the rationale for separate analyses of the 2002–2014 and 2010–2018 datasets.

## Statistical analysis

To explore the plausible drivers of RSV seasonal epidemics, we applied two analytical approaches. The first involved using a Generalized Additive Model (GAM), whilst the second involved a time-series-susceptible-infected-recovered model (TSIR). GAMs are generalized linear models with predictors specified as smooth functions of the covariates [31]. The cubic spline smoothing function was adopted as it has been shown to yield smoother interpolations when compared to other methods [31,32]. The model employed a negative binomial with a log-link function due to over-dispersion in the weekly number of RSV cases. The model included meteorological variables, weekly number of births, school term (in or out of term time), and RSV activity in the previous 3 weeks (due to the serial correlation resulting from the infection dynamics of RSV). We explored models for the lagged variables up to the previous one week (i.e. values observed in the previous week for the covariates) to explore delay effects.

A cubic smoothing spline function of time was included to control for seasonal patterns and long-term trend. Statistical significance was determined by an alpha level of 5%. All analyses were done using the mgcv package in the R statistical software [33].

The TSIR method [34–36] was applied to enhance the exploration of how changes in birth rates affect RSV dynamics as it has been shown for measles epidemics in England and Wales [36]. Furthermore, it also provided an alternative approach to explore the plausible drivers of RSV transmission dynamics in this region [34]. This discrete time epidemic model comprises two state variables, namely: the infected and the susceptible. Since the susceptible is unobserved, we reconstructed it using a set of recursive difference equations utilizing the weekly number of RSV cases, weekly number of births and interpolated total weekly population over the years using the tsiR package implemented in R statistical software [35]. Next, using the reconstructed susceptible population over time, we reconstructed an empirical estimate of the transmission rate which was subsequently used as a dependent variable in both linear regression and generalized additive negative binomial models with meteorological variables and school term data as independent variables. More information on the TSIR and GAM methodologies is contained in the supplementary material.

Both analyses were performed on each of the five imputed 2002–2014 datasets and the five imputed 2010–2018 datasets obtained as described above in the data pre-processing sub-section. Our choice of independent analysis was motivated by the fact that imputation of the missing data in the meteorological variables was performed at daily resolutions and the weekly averages were obtained to merge with the other considered datasets. Robust imputation methodologies for data collected in different resolutions are still lacking in the literature and thus future work in this area is warranted.

## Results

In all our analyses, we observed that the weekly number of births was only significant at lag two and three (number of births observed in the previous two and three weeks, respectively). Hence in the GAM models, we considered a lag of two weeks of the weekly number of births to coincide with the meteorological variables and school term, and a lag of three weeks of the weekly births to coincide with a lag of one week of the meteorological variables and school term. The two approaches (GAM and TSIR) yielded similar results, hence, we report the GAM analyses below, the TSIR results are in the supplementary material.

In the 2002–2014 datasets, we observed that maximum temperature both at lags zero and one week was a significant predictor of the weekly number of RSV cases (p-value <0.05, Table 1). Absolute humidity was a significant predictor at lag one week (p-value <0.05, Table 1). The weekly number of births was a significant predictor of the weekly number of RSV cases at both lags two and three weeks (p-value <0.05, Table 1). Rainfall and school terms did not yield significant relationships. The plots of the smooth terms of maximum temperature, absolute humidity and weekly number of births indicated non-linear relationships with the weekly number of RSV cases (S1 Fig).

Analyses using the 2010–2018 datasets yielded similar results where maximum temperature and absolute humidity were significant predictors of the weekly RSV cases at both lags zero and one week (p-value <0.05, S1 Table in S1 File). Similarly, weekly number of births at both lags two and three weeks was a significant predictor of the RSV cases (S1 Table in S1 File). The plots of the smooth terms of maximum temperature, absolute humidity, and weekly number of births showed non-linear relationships with the weekly number of RSV cases (Fig 2).

Generally, the correlation coefficients between the observed and the predicted number of weekly RSV cases was high for both the 2002–2014 datasets (Table 1), and the 2010–2018

**Table 1. Generalized additive negative binomial models with lags zero and one week of meteorological variables and lags two and three weeks of weekly number of births, respectively, in the 2002–2014 datasets.**

| Imputed datasets | Smooth terms EDF (p-value) † | | | | | Adj.R$^2$ | % Dev. explained | Pred. Corr. Coef ‡ |
|---|---|---|---|---|---|---|---|---|
| | Maximum Temperature (Lag zero) | Absolute Humidity (Lag zero) | Weekly Births (Lag two) | Rainfall (Lag zero) | School Term (Lag zero) | | | |
| Data 1 | 2.841 (< 0.001) | 0.0014 (0.588) | 2.245 (0.001) | 0.833 (0.143) | -0.091 (0.330) | 0.575 | 64.9 | 0.774 |
| Data 2 | 2.786 (< 0.001) | 0.001 (0.578) | 2.280 (0.001) | 0.794 (0.176) | -0.092 (0.328) | 0.572 | 64.8 | 0.773 |
| Data 3 | 3.195 (< 0.001) | 0.0005 (0.639) | 2.367 (<0.001) | 0.0007 (0.492) | -0.071 (0.445) | 0.566 | 65.0 | 0.770 |
| Data 4 | 2.910 (< 0.001) | 0.002 (0.640) | 2.274 (<0.001) | 0.671 (0.178) | -0.074 (0.429) | 0.572 | 65.2 | 0.774 |
| Data 5 | 3.052 (< 0.001) | 0.001 (0.724) | 2.303 (<0.001) | 0.011 (0.347) | -0.079 (0.396) | 0.571 | 65.0 | 0.733 |

Lag one of meteorological variables and school term

| Imputed datasets | Maximum Temperature (Lag one) | Absolute Humidity (Lag one) | Weekly Births (Lag three) | Rainfall (Lag one) | School Term (Lag one) | Adj.R$^2$ | % Dev. explained | Pred. Corr. Coef‡ |
|---|---|---|---|---|---|---|---|---|
| Data 1 | 2.120 (0.007) | 2.687 (0.015) | 2.265 (<0.001) | 1.126 (0.061) | -0.068 (0.461) | 0.602 | 65.7 | 0.791 |
| Data 2 | 2.262 (0.005) | 2.127 (0.014) | 2.30 (<0.001) | 0.450 (0.229) | -0.063 (0.497) | 0.595 | 65.3 | 0.787 |
| Data 3 | 2.380 (< 0.001) | 1.929 (0.046) | 2.287 (<0.001) | 0.695 (0.170) | -0.055 (0.557) | 0.595 | 65.4 | 0.787 |
| Data 4 | 2.215 (0.003) | 2.110 (0.051) | 2.229 (<0.001) | 0.846 (0.141) | -0.061 (0.510) | 0.599 | 65.2 | 0.789 |
| Data 5 | 2.398 (< 0.001) | 1.784 (0.049) | 2.293 (<0.001) | 0.553 (0.239) | -0.051 (0.580) | 0.589 | 65.6 | 0.784 |

†EDF: Effective degrees of freedom for the estimated smooth terms.

‡ Correlation coefficient between the estimated and observed RSV.

datasets (S1 Table in S1 File). The curves of the predicted and the observed weekly RSV cases closely followed each other for both the 2002–2014 datasets (Fig 3), and in the 2010–2018 datasets (Fig 4). The percentage of deviance explained was relatively higher for models including the lagged values of meteorological variables, weekly number of births and school terms (Table 1 & S1 Table in S1 File). The 2010–2018 datasets yielded higher values of the percentages of the deviance explained, the adjusted R-squared, and the correlation coefficients between the observed and predicted weekly number of RSV cases (Table 1 & S1 Table in S1 File).

## Discussion

In this study, we found that maximum temperature, absolute humidity and weekly number of births are significant predictors of the weekly number of RSV cases from the independent analysis of ten imputed datasets.

Although humans can reproduce throughout the year, some human populations exhibit seasonal variation in reproduction leading to seasonal birth rates/patterns [37]. These variations can be utilized to characterize the seasonal variations in the incidence of infectious diseases in early childhood [20,37]. In our study, the weekly number of births was a significant

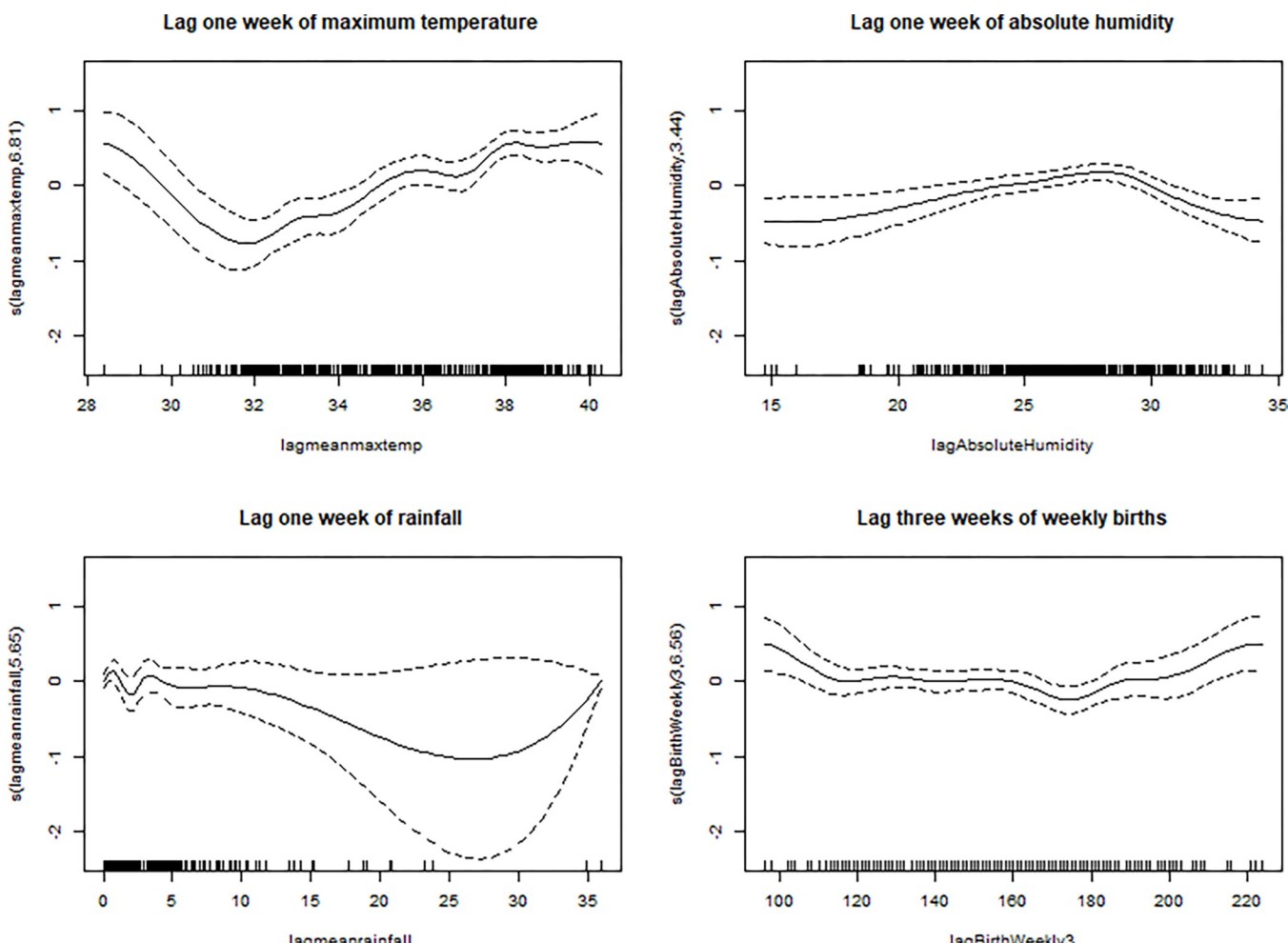

**Fig 2. Plots of the meteorological smooth terms in the first imputed 2010–2018 dataset.** The dashed lines are the 95% confidence intervals.

predictor of RSV. Seasonality of births leads to pulses of new hosts in the population leading to an expansion of the susceptible population, a significant factor modulating the transmission dynamics of childhood infectious diseases [19,20]. A study in the US [38] found out that year-to-year and state-to-state variations in birth rates determined the timing of rotavirus epidemics. Another study in the US [39] showed variations in RSV epidemic timing where counties with larger population sizes had earlier epidemic peaks and recorded higher RSV incidence. A mathematical modelling study [40] found that changes in birth rates led to either annual or biennial RSV seasonal epidemics. Another modelling study [16] in the US showed that the transition to annual seasonal RSV epidemics in 2000 from biennial patterns in 1990s in California could have been modulated by the changes in the birth rates. Similarly, another study [20] in Sub-Saharan Africa showed that changes in seasonal birth rates can either lead to annual or biennial dynamics of infectious diseases occurring early in life such as measles. Thus, the observed associations between the weekly number of births and weekly RSV cases in this region could suggest that the seasonal changes in births could be a potential driver of the observed RSV seasonal epidemics. Further research is needed in this area.

School term did not yield significant relationships in our study. This might suggest that school terms do not play a major role in determining the RSV seasonality in this region.

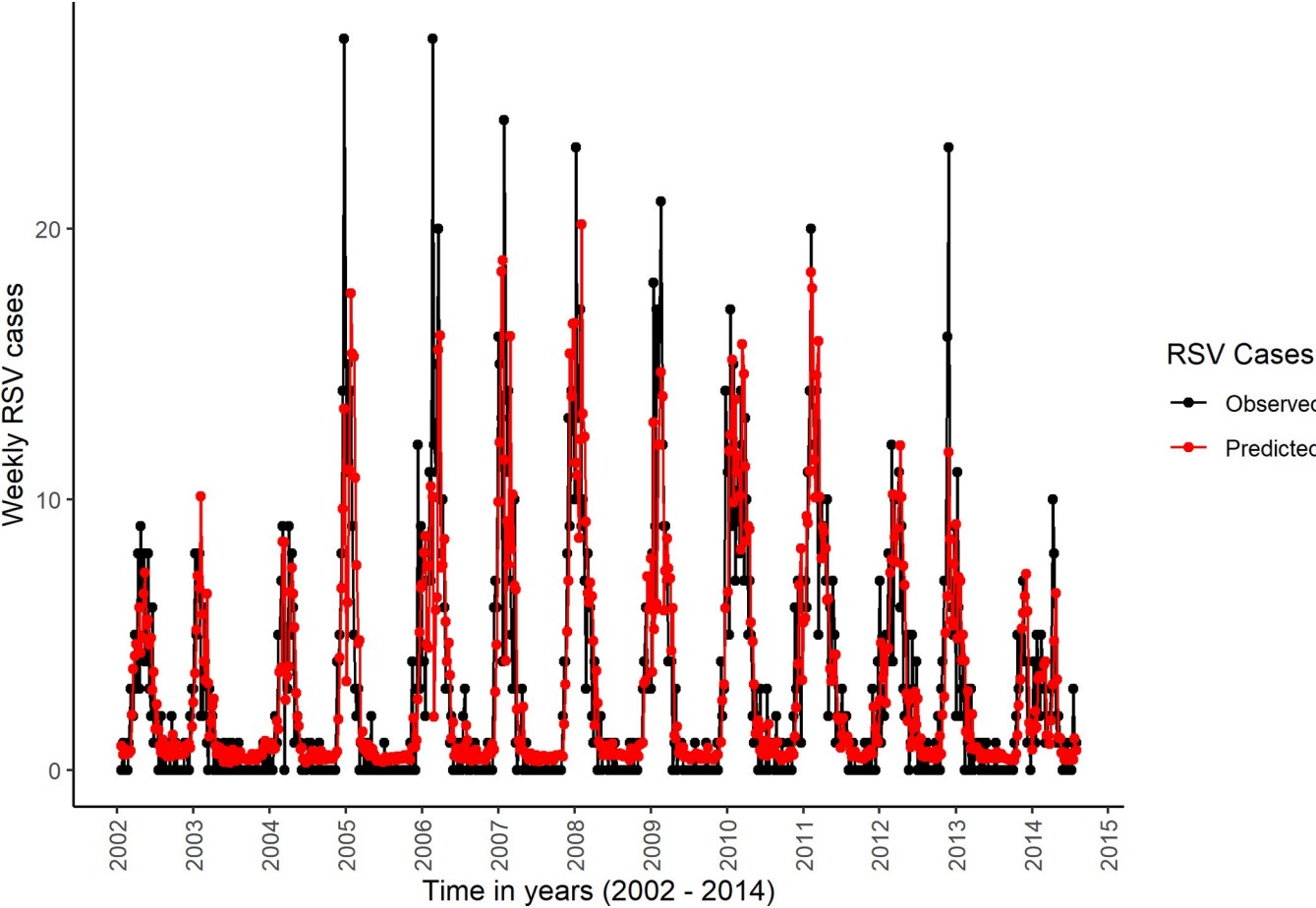

**Fig 3. Plot of observed and predicted weekly RSV cases in the first imputed 2002–2014 dataset.** The black line connects the observed weekly RSV cases while the red line connects the predicted weekly RSV cases.

However, aggregation of children in schools has been found to play an important role in increasing the transmission rates of measles in England and Wales [41], Influenza in England and varicella zoster virus in several settings [42]. For RSV, it has been hypothesized that increased contact rates during school terms could play a crucial role in influencing its seasonality [43]. In addition, several studies have demonstrated that RSV is mostly introduced into the households by school going siblings consequently leading to the infections to the infants [44,45]. However, results from 4 recent studies do not appear to support this earlier hypothesis [14–16,34] consistent with what we observed in our study. Insights into why 'school terms' is not identified as a significant contributor to RSV seasonality would be gained from further studies of RSV transmission patterns in the school setting, including linking RSV individual-level infections data and social contact patterns for school going children at school-level, community and household levels to gain deeper understanding of 'who infects who' in the community.

Rainfall did not yield significant relationships in our study implying that it might not be a plausible driver of RSV seasonal epidemics in this region. This is in contrast with several other studies that have found associations between rainfall and RSV activity in the following regions: Baguio city in the Philippines, Okinawa in Japan [46], tropical Kolkata in India [47], and some states in the US [16], Lombok island in Indonesia [12], Kuala Lumpur in Malaysia [47], the

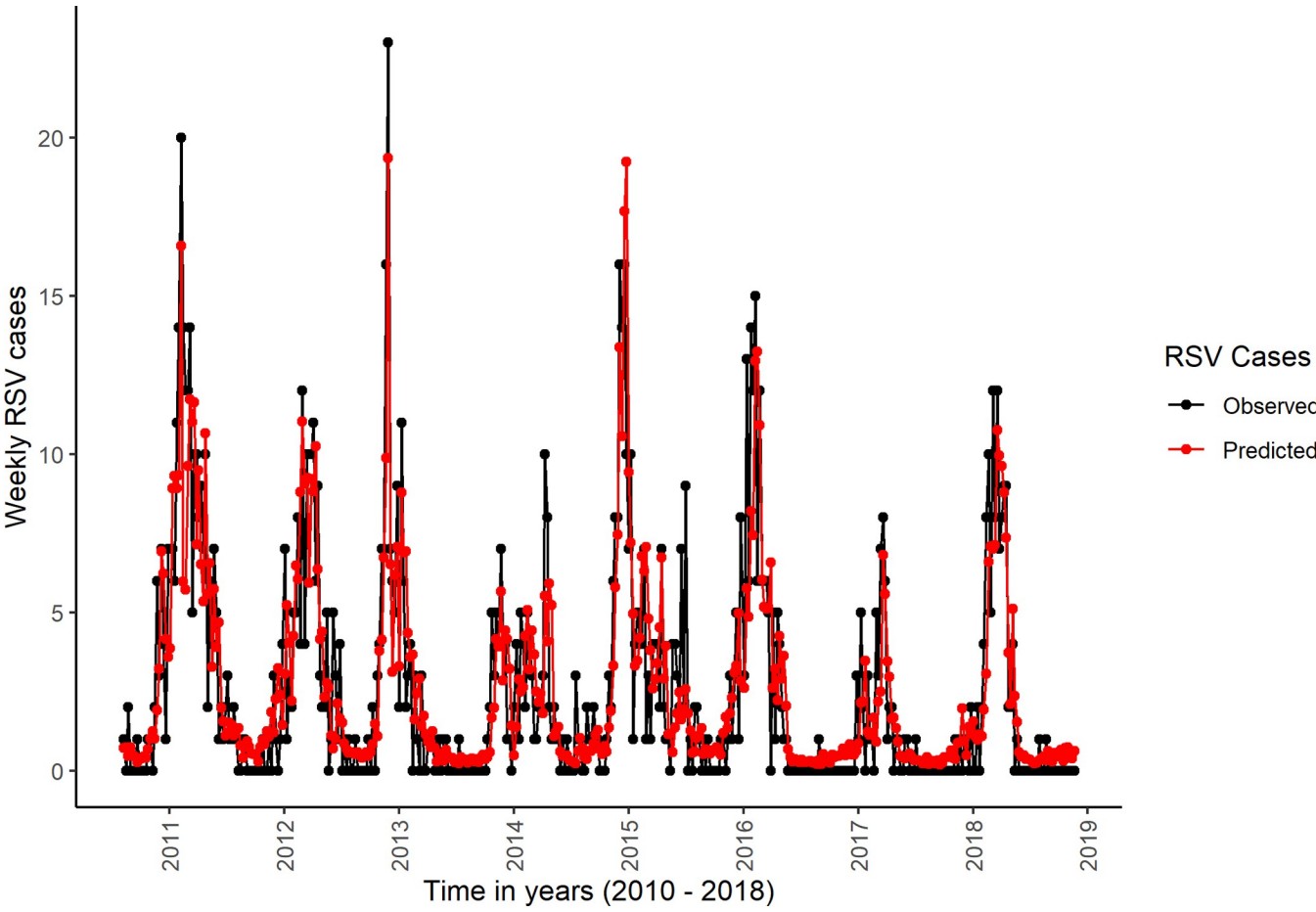

**Fig 4. Plot of observed and predicted weekly RSV cases in the first imputed 2010–2018 dataset.** The black line connects the observed weekly RSV cases while the red line connects the predicted weekly RSV cases.

Philippines [15], Gambia in West Africa [17], and an Equatorial city in Brazil [48]. Maximum temperature showed non-linear relationship with RSV activity in this region. Interestingly, absolute humidity also exhibited a non-linear relationship with the RSV activity. A previous laboratory study [49] conducted to understand the stability of RSV in varying conditions found out that at 37°C, RSV inactivation took place over a period of 3 days. Varying the temperature to 4°C, indicated that 99% inactivation required a duration of 6 days or more. Another study [50] at a constant temperature of 20.5°C, the maximal stability of RSV was recorded at 80–90% humidity, and maximal inactivation occurred at 20–30% humidity. Thus, we postulate that stability of RSV in large particle aerosols in this tropical setting could be dependent on variations in maximum temperature and absolute humidity. This could presumably modulate aerosol transmission through altering the stability of RSV in the environment.

While meteorological factors and weekly number of births could likely influence RSV seasonality, other plausible drivers of RSV epidemics such as malnutrition [22], household crowding and population density [23] could be explored. In our study, we used outdoor measurements for the meteorological variables which could differ with indoor conditions where the study domain (children under 5 years) spend much of their time. School term data was inferred from the 2019 academic calendar, there might be year to year variations over the study period. Our study assumed that the meteorological data had the same spatial coverage in

all locations where the study domain lived and there was no spatial heterogeneity based on the distribution of these RSV cases. Furthermore, although Kenya sits on the equator, it has quite different climatic regions. Thus, the chosen study location of coastal Kenya may differ from other regions and hence our results might not be generalizable. More studies, using data from geographically distinct locations with variation in the seasonal patterns and potential drivers are needed in this regard.

## Conclusion

Our study has identified maximum temperature, absolute humidity and weekly number of births as potential drivers of RSV epidemics in this region. These findings contribute to the current understanding of the potential drivers of RSV seasonality in a tropical low-income setting where little is known about the plausible drivers of the RSV seasonality. Identifying these factors is important in improving the understanding of the transmission dynamics of the virus in this setting. Furthermore, these factors can be utilized in the formulation of prediction models of RSV activity in this setting. This potentially could provide early warnings of RSV seasons and subsequently inform the timings of RSV immunizations using vaccine candidates currently in the pipeline. However, more research is needed in other regions with different climatic classifications to shed more light on the observed relationships.

## Supporting information

**S1 Checklist. Completed questionnaire on inclusivity in global research.**
(DOCX)

**S1 Fig. Plots of the meteorological smooth terms in the first imputed 2002–2014 dataset.** The dashed lines are the 95% confidence intervals.
(TIF)

**S2 Fig. The estimated transmission rate from the time-series-susceptible-infected-recovered (TSIR) model with the 95% confidence interval.**
(TIF)

**S3 Fig. Plot of the mean of 100 stochastic simulations from the TSIR model fit (red) plotted alongside observed data of weekly RSV cases (blue).**
(TIF)

**S4 Fig. Plot for the weekly number of RSV cases and the weekly number of births from the year 2002 to 2018.**
(TIF)

**S5 Fig. Weekly number of RSV cases compared to mean weekly maximum temperature, absolute humidity, rainfall, and potential evapotranspiration between the year 2002 and 2014.**
(TIF)

**S6 Fig. Weekly number of RSV cases compared to mean weekly maximum temperature, absolute humidity, rainfall, and potential evapotranspiration between the year 2010–2018.**
(TIF)

**S1 File. Details on data pre-processing and Time Series Susceptible-Infected-Recovered (TSIR) methodology, and supplementary results.**
(DOCX)

## Acknowledgments

This study is published with permission from the Director of KEMRI.

## Author Contributions

**Conceptualization:** James Wambua, D. James Nokes, Niel Hens.

**Data curation:** James Wambua, Nickson Murunga.

**Formal analysis:** James Wambua.

**Funding acquisition:** D. James Nokes.

**Methodology:** James Wambua, Niel Hens.

**Software:** James Wambua.

**Supervision:** D. James Nokes, Niel Hens.

**Visualization:** James Wambua.

**Writing – original draft:** James Wambua.

**Writing – review & editing:** James Wambua, Patrick K. Munywoki, Pietro Coletti, Bryan O. Nyawanda, Nickson Murunga, D. James Nokes, Niel Hens.

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
