## [Decision Letter · Decision Letter 0]

5 Sep 2022

PONE-D-22-07030Drivers of respiratory syncytial virus seasonal epidemics in children under 5 years in Kilifi, coastal KenyaPLOS ONE

Dear Dr. Wambua,

Thank you for submitting your manuscript to PLOS ONE. After careful consideration, we feel that it has merit but does not fully meet PLOS ONE’s publication criteria as it currently stands. Therefore, we invite you to submit a revised version of the manuscript that addresses the points raised during the review process.

We look forward to receiving your revised manuscript.

Kind regards,

Emanuele Giorgi

Academic Editor

PLOS ONE

3. You indicated that you had ethical approval for your study. In your Methods section, please ensure you have also stated whether you obtained consent from parents or guardians of the minors included in the study or whether the research ethics committee or IRB specifically waived the need for their consent.

4.We note that the grant information you provided in the ‘Funding Information’ and ‘Financial Disclosure’ sections do not match. When you resubmit, please ensure that you provide the correct grant numbers for the awards you received for your study in the ‘Funding Information’ section.

5. We note that [Figure 1] in your submission contain [map/satellite] images which may be copyrighted. All PLOS content is published under the Creative Commons Attribution License (CC BY 4.0), which means that the manuscript, images, and Supporting Information files will be freely available online, and any third party is permitted to access, download, copy, distribute, and use these materials in any way, even commercially, with proper attribution. For these reasons, we cannot publish previously copyrighted maps or satellite images created using proprietary data, such as Google software (Google Maps, Street View, and Earth). For more information, see our copyright guidelines: http://journals.plos.org/plosone/s/licenses-and-copyright.

a. You may seek permission from the original copyright holder of Figure(s) [#] to publish the content specifically under the CC BY 4.0 license. 

Natural Earth (public domain): http://www.naturalearthdata.com/.

Reviewers' comments:

Reviewer's Responses to Questions

**Comments to the Author**

1. Is the manuscript technically sound, and do the data support the conclusions?

Reviewer #1: Yes

Reviewer #2: Yes

2. Has the statistical analysis been performed appropriately and rigorously? 

Reviewer #1: I Don't Know

Reviewer #2: Yes

3. Have the authors made all data underlying the findings in their manuscript fully available?

Reviewer #1: Yes

Reviewer #2: Yes

4. Is the manuscript presented in an intelligible fashion and written in standard English?

Reviewer #1: Yes

Reviewer #2: Yes

5. Review Comments to the Author

Reviewer #1: Thank you for submitting this valuable manuscript because it is the one of the most important topics to improve survival of children particularly in Sub-Saharan Africa. My comments and questions are as follows:

1. Introduction: In your introduction part, you stated a statement "Two maternal booster F protein vaccines are in Phase III trials", please site a reference.

2. Methods: kindly give the study design even though you have used secondary data. which GAM fitting method did you used? Backfilling algorithm or else?? please give some explanation about it. Again, in your methods please give a description about model comparison methods used. As you know overfitting is one problem that can be encountered in GAM. So, how do you assess the overfitting issue?? Kindly give explanation about it.

3. Conclusion: you stated that school term is not significant predictor of seasonality of RSV, and you simply compare and contrast with findings of other previous studies please discuss the possible justification about it.

Reviewer #2: Tables: Table 1 is not captioned

DISCUSSION: Be specific on maximum temperature, absolute humidity and weekly number of births. There should not be sub-headings under discussion

REFERENCES: In Vancouver reference style, it is only after 6 authors and their initials before ‘’et al’’ can be used (inserted)

CONTRIBUTION: The authors did not state the contribution of this study to the existing body of knowledge.

STUDY RESULTS IMPLICATION : Not stated

6. PLOS authors have the option to publish the peer review history of their article (what does this mean?). If published, this will include your full peer review and any attached files.

Reviewer #1: **Yes: **Bayley Adane Takele

Reviewer #2: **Yes: **Prof Abdulraheem Ibraheem

---

## [Author Response · Author response to Decision Letter 0]

19 Oct 2022

Reviewer 1 Comments:

Thank you for submitting this valuable manuscript because it is the one of the most important topics to improve survival of children particularly in Sub-Saharan Africa. My comments and questions are as follows:

We thank the reviewer for the appreciation of our work and for taking their time to review it and for their thoughtful comments and suggestions.

Comments:

1. Introduction: In your introduction part, you stated a statement "Two maternal booster F protein vaccines are in Phase III trials", please site a reference.

Thank you for pointing out this missing reference in our work. We have included a reference as suggested. The citation is entitled “Gunatilaka A, Giles ML. Maternal RSV vaccine development. Where to from here?. Human Vaccines & Immunotherapeutics. 2021 Nov 2;17(11):4542-8.”

2. Methods: kindly give the study design even though you have used secondary data.

This is an interesting suggestion. We have added a statement in the methods that indicate the study design for the pre-existing data. The statement reads, “Data used were collected from routine longitudinal hospital case surveillance for RSV, and routine longitudinal surveillance for the number of births and meteorological variables.”

which GAM fitting method did you used? Backfilling algorithm or else?? please give some explanation about it. 

We used the local scoring algorithm, which iteratively fits weighted additive models by the backfitting algorithm as implemented in the mgcv package in R (1). The algorithm estimates each smooth component of an additive model by iteratively smoothing partial residuals, with respect to the covariates that the smooth relates to. The basic idea is that the partial residuals relating to the jth smooth term are the residuals resulting from subtracting all the current model term estimates from the response variable, except the estimate of the jth smooth term. This algorithm has the advantage of flexibility since it can choose from a wide variety of smoothing methods for component estimation (2).

in your methods please give a description about model comparison methods used.

We used both the Akaike Information Criterion (AIC) and Generalized Cross Validation (GCV) scores to compare different models. 

As you know overfitting is one problem that can be encountered in GAM. So, how do you assess the overfitting issue?? Kindly give explanation about it.

We thank the reviewer for pointing this out. Indeed overfitting is one of the key problems that can be encountered when using the generalized additive models (GAM). Since the GAMs are simply generalized linear models estimated subject to smoothing penalties of the covariates, the most substantial difficulty is choosing the degree of penalization for the smoothing parameters which to a greater extend controls the overfitting. In our study, we used the package “mgcv” in R as indicated in the statistical analysis section and applied the Generalized Cross Validation (GCV) smoothing parameter estimation. Furthermore, we choose a value of 1.5 for the gamma parameter. This parameter allows one to increase the penalty on each models degrees of freedom in order to increasingly produce more smooth models. It has been demonstrated that a gamma of around 1.4 can be a sensible choice for suppressing over-fitting without much degradation in prediction error performance (3). 

To assess whether there was presence of overfitting, we generated plots of the residual model deviances and investigated whether there was autocorrelation in these residuals. 

Since we used two sets of time series datasets, i.e, 2002-2014 and 2010-2018, and considered five imputed datasets for each dataset, these datasets also acted as validation sets to each other. Fitted models in the imputed datasets yielded similar results and comparable adjusted R-Squared and the deviance explained further indicating no presence of over-fitting. 

In order to provide information on these methodological clarifications in our work while keeping the statistical analysis section easy to follow, we created a section called “Supplementary information on Generalized Additive Models” in the Supplementary material and added the following information:

We employed the backfitting algorithm as implemented in the ‘mgcv’ package in R. This algorithm estimates each smooth component of additive model by iteratively smoothing partial residuals with respect to the covariates that the smooth relates to. The models were compared using both the Akaike Information Criterion (AIC) and the Generalized Cross Validation (GCV) scores. Given the inherent problem of over-fitting in the generalized additive models, we set the value of gamma parameter to be 1.5. This parameter allows one to increase the penalty on each model’s degrees of freedom in order to increasingly produce more smooth models, as over-fitting is controlled by the degree of penalization. We then inspected the plots of the residual model deviances to see whether there were autocorrelations in the residuals which could indicate possible presence of over-fitting in the models. 

3. Conclusion: you stated that school term is not significant predictor of seasonality of RSV, and you simply compare and contrast with findings of other previous studies please discuss the possible justification about it.

This is an interesting point. The transmission dynamics of RSV in school aged children and the role of school settings are poorly understood. It remains uncertain as to whether the main driver of RSV transmission is primary infections or reinfections (4), if the former, and since practically all individuals experience a first infection by the third year of life (5), then school aged children would not be of importance. We conducted a surveillance study in a rural school in Kenya and identified very little RSV infection supporting the notion of little importance of RSV in the school setting (6). However, there are few studies on the epidemiology of RSV in school setting and further work in this area is warranted. 

Based on the reviewer’s suggestion to provide a possible justification of why school terms don’t seem to play a crucial role, we add the following lines in the discussion section highlighting that more research needs to be done to shed light on the role of schools on RSV seasonality.

Insights into why ‘school terms’ is not identified as a significant contributor to RSV seasonality would be gained from further studies of RSV transmission patterns in the school setting, including linking RSV individual-level infections data and social contact patterns for school going children at school-level, community and household levels to gain deeper understanding of ‘who infects who’ in the community.

Reviewer #2: 

1. Tables: Table 1 is not captioned

We thank the reviewer for pointing out this. The caption of the table is now included above the table (lines 231-233) and reads “Table 1. Generalized additive negative binomial models with lags zero and one week of meteorological variables and lags two and three weeks of weekly number of births, respectively, in the 2002 - 2014 datasets.”

2. DISCUSSION: Be specific on maximum temperature, absolute humidity and weekly number of births. There should not be sub-headings under discussion

We have now explicitly referred to maximum temperature and absolute humidity where this was not specific. We have also removed the sub-headings in the discussion section as per the reviewers suggestion.

3. REFERENCES: In Vancouver reference style, it is only after 6 authors and their initials before ‘’et al’’ can be used (inserted)

We have changed the reference style to Vancouver and now we have six authors and their initials before et al. 

4. CONTRIBUTION: The authors did not state the contribution of this study to the existing body of knowledge.

This point is well noted. We have added the contribution of the study into the existing body of knowledge in the conclusion section and reads as follows: “These findings contribute to the current understanding of the potential drivers of RSV seasonality in a tropical low-income setting where little is known about the plausible drivers of RSV seasonality.”

5. STUDY RESULTS IMPLICATION : Not stated

We thank the reviewer for this comment. We have added the implication of the study results in the conclusion section and reads as follows. “Furthermore, these factors can be utilized in the formulation of prediction models of RSV activity in this setting. This potentially could provide early warnings of RSV seasons and subsequently inform the timings of RSV immunizations using vaccine candidates currently in the pipeline”.

---

## [Editor Report · Decision Letter 1]

9 Nov 2022

Drivers of respiratory syncytial virus seasonal epidemics in children under 5 years in Kilifi, coastal Kenya

PONE-D-22-07030R1

Dear Dr. Wambua,

We’re pleased to inform you that your manuscript has been judged scientifically suitable for publication and will be formally accepted for publication once it meets all outstanding technical requirements.

Kind regards,

Emanuele Giorgi

Academic Editor

PLOS ONE
---

## [Editor Report · Acceptance letter]

16 Nov 2022

PONE-D-22-07030R1 

Drivers of respiratory syncytial virus seasonal epidemics in children under 5 years in Kilifi, coastal Kenya 

Dear Dr. Wambua:

I'm pleased to inform you that your manuscript has been deemed suitable for publication in PLOS ONE. Congratulations! Your manuscript is now with our production department. 

Kind regards, 

on behalf of

Dr. Emanuele Giorgi 

Academic Editor

PLOS ONE